# Is Chronic Varicocele a Risk Factor for Secondary Hyperparathyroidism?

**DOI:** 10.3390/jcm11030716

**Published:** 2022-01-28

**Authors:** Rossella Cannarella, Rosita A. Condorelli, Sarah Perelli, Aldo E. Calogero, Emanuela Greco, Antonio Aversa, Sandro La Vignera

**Affiliations:** 1Department of Clinical and Experimental Medicine, University of Catania, 95123 Catania, Italy; rossella.cannarella@phd.unict.it (R.C.); rosita.condorelli@unict.it (R.A.C.); sarah.perelli@libero.it (S.P.); acaloger@unict.it (A.E.C.); 2Department of Health Sciences, University “Magna Graecia”, 88100 Catanzaro, Italy; emanuela.greco@unicz.it; 3Department of Experimental and Clinical Medicine, University “Magna Graecia”, 88100 Catanzaro, Italy; aversa@unicz.it

**Keywords:** 25-hydroxylase, varicocele, vitamin D, parathyroid hormone, testicular function

## Abstract

Objective: To assess whether varicocele affects testicular 25-hydroxylase activity. Methods: Twenty normozoospermic patients with bilateral varicocele (grade III according to the Dubin and Amelar classification) without indications to undergo varicocele repair (normal sperm parameters and testicular volume; no scrotal pain) were consecutively enrolled and followed-up for four years. Serum levels of parathyroid hormone (PTH), calcium, and 25-hydroxy-cholecalciferol [25(OH)D] along with serum luteinizing hormone (LH), follicle-stimulating hormone (FSH), total testosterone (TT), conventional sperm parameters, sperm DNA fragmentation (SDF) rate, and testicular volume (TV) were measured annually for three years. PTH, calcium, and 25(OH)D serum levels over time were compared with those of age- and body mass index (BMI)-matched control group of twenty varicocelectomized patients. Main results: Both intra- and between-group analyses showed that serum PTH levels increased significantly over time in parallel with a significant decline in 25(OH)D levels. Serum calcium levels did not change significantly. At the same time, signs of mild Leydig and Sertoli cell dysfunction were found, such as an increase in gonadotropins and decreased TT and VT. However, conventional sperm parameters and SDF rate did not change significantly. Conclusion: This prospective controlled study provides the first evidence of a negative impact of bilateral grade III varicocele on testicular 25-hydroxylase activity. Accordingly, the patients included in this study showed a significant increase in PTH and a decrease in 25(OH)D levels over time. Patients with varicocele deserve endocrinologic counseling.

## 1. Introduction

Testicular varicocele is an abnormal dilation and tortuosity of the pampiniform plexus that drains the testis. Its prevalence has been estimated to be up to 15% in adulthood. It is higher in infertile patients who have a prevalence of 19–41% in primary infertility and up to 80% in secondary infertility [1]. Clinical staging of varicocele severity can be done using the Dubin and Amelar classification. It consists of four grades: grade 0 which identifies the subclinical varicocele (not clinically detectable but diagnosed by ultrasound); grade I, palpable only during the Valsalva maneuver; grade II, appreciable even without Valsalva maneuver; grade III, visible on inspection [2].

Several lines of evidence show a negative role of varicocele on testicular function. Indeed, patients with varicocele have lower sperm quality and pregnancy outcomes compared to healthy controls [3,4,5]. Furthermore, meta-analytic studies have shown its negative influence on Leydig cell function and, therefore, on testosterone biosynthesis [6].

The exact mechanisms by which varicocele damage the testicular function are not entirely clear. Several theories have been proposed. First of all, varicocele causes scrotal hyperthermia that negatively impacts spermatogenesis. Furthermore, a decreased heat-shock protein (HSP) expression may contribute to heat stress that, in turn, increases oxidative stress (OS) and apoptosis. Moreover, blood stasis in varicose veins can promote the entrapment of leucocytes that release reactive oxygen species (ROS). Finally, varicocele can cause testicular hypoxia. Reduced expression of anti-apoptotic genes and increased expression of pro-apoptotic genes increases the susceptibility to OS and thermogenic damage. The different degrees of susceptibility may explain why varicocele, even of a severe degree, is not always associated with testicular damage [7].

So far, the effects of varicocele on testicular 25-hydroxylase activity have not been evaluated. This microsomal enzyme, codified by the *CYP2R1* gene, has been identified in human Leydig cells [8]. It catalyzes the hydroxylation of cholecalciferol into 25-hydroxy-cholecalciferol [25(OH)D]. This is further hydroxylated into 1,25(OH_2_)-vitamin D3 (calcitriol) in the kidney under the stimulation of parathyroid hormone (PTH). Calcitriol, the active form of vitamin D, stimulates the absorption of calcium from the kidney tubules and bowel and favors bone mineralization. Several other genomic and non-genomic effects of calcitriol have been reported in many tissues [9].

In healthy testis, the expression of 25-hydroxylase is higher than that reported in the liver. However, its expression is reduced in testis with hypospermatogenesis or Sertoli cell-only syndrome (SCOS), and this predisposes to 25(OH)D deficiency [8]. Furthermore, patients with late-onset hypogonadism are at risk of developing 25(OH)D deficiency and secondary hyperparathyroidism. This has been shown by studies investigating the best therapeutic scheme for Vitamin D replacement, able to normalize both 25(OH)D and serum parathyroid hormone (PTH) levels in these patients [10]. These findings suggest that patients with diseases that lead to testicular dysfunction should not only undergo clinical management for abnormalities of sperm parameters or decreased testosterone biosynthesis but also serum 25(OH)D levels resulting from Leydig cell dysfunction.

Whether varicocele can affect the activity of 25-hydroxylase in Leydig cells is not known. Therefore, this study aimed to evaluate this topic. To accomplish this, twenty consecutive normozoospermic patients with bilateral varicocele of grade III (according to the Dubin and Amelar scale) were prospectively evaluated for PTH, calcium, and 25(OH)D (primary outcomes) for three years. PTH, calcium, and 25(OH)D serum levels over time were compared with those of age- and body mass index (BMI)-matched control group of twenty varicocelectomized patients. The secondary outcomes were serum gonadotropin and total testosterone (TT) levels, conventional sperm parameters, sperm DNA fragmentation (SDF) rate, and testicular volume (TV).

## 2. Subjects and Methods

### 2.1. Patient Selection

This is a prospective study performed in men who referred to the Division of Endocrinology, Metabolic Diseases and Nutrition, University of Catania, for varicocele, from March 2020. We selected the first 20 patients with specific clinical features. The inclusion criteria were: bilateral varicocele of grade III according to the Dubin and Amelar scale [2] and without the following indications to varicocele repair: normal sperm parameters, no scrotal pain, and TV in the normal range (>15 mL) by scrotal ultrasound. Furthermore, the selected patients did not have a head injury, endocrine disorders (hypogonadism, hyperprolactinemia, Cushing syndrome, acromegaly, hypopituitarism), elevated serum follicle-stimulating hormone (FSH) levels, systemic diseases (kidney and/or liver diseases), and genetic disorders. Finally, past history of hyperparathyroidism, hypercalcemia, and vitamin D supplementation at enrolment or during the study were considered exclusion criteria. Forty Caucasian patients were enrolled. Twenty did not undergo varicocele repair (Patient group) and twenty underwent varicocele repair (Control group). The patient group had a mean (± SD) age of 29.9 ± 6.2 years, whereas the Control group had a mean (±SD) age of 31.0 ± 4.0 years (*p* > 0.1). The two groups were also BMI-matched (Patients: 22.6 ± 1.4 kg/m^2^; Controls: 23.5 ± 2.0 kg/m^2^; *p* > 0.1). Each endpoint was measured at enrollment (T0) and annually for three years (T1, T2, and T3). This consisted of PTH, calcium, and 25(OH)D serum levels as primary outcomes. Serum luteinizing hormone (LH), FSH, and TT levels, conventional sperm parameters, SDF rate, and TV were evaluated as secondary outcomes.

### 2.2. Hormonal Measurements

Blood testing was performed in fasting conditions, at 8:00–9:00 am. The serum levels of LH, FSH, TT, PTH, calcium, and 25(OH)D were measured by electrochemiluminescence (Hitachi-Roche equipment, Cobas 6000, Roche Diagnostics, Indianapolis, IN, USA). The reference values were as follows: LH 1.14–8.75 IU/L, FSH 0.95–11.95 IU/L, TT 0.478–9.8 ng/mL, PTH 0.02–1.23 ng/mL, calcium 8.8–10.6 mg/dL, 25(OH)D 1–100 ng/mL.

### 2.3. Sperm Analysis

Semen samples were collected by masturbation into a sterile container after 2–7 days of sexual abstinence and were analyzed immediately after liquefaction. The analysis was conducted according to the 2010 WHO guidelines. Each sample was evaluated for seminal volume, pH, sperm count, progressive motility, morphology, and round cell concentration [11].

### 2.4. Sperm DNA Fragmentation

SDF was measured through flow cytometry by terminal deoxynucleotidyl transferase-mediated deoxyuridine triphosphate nick-end labeling (TUNEL) staining, using an EPICS XL (Becker Coulter, Milan, Italy). The negative control was obtained by not adding terminal deoxynucleotidyl transferase to the reaction mix, while the positive control was obtained by pretreating spermatozoa with 1 mg/mL of RNase-free deoxyribonuclease I (Sigma Chemical, St. Louis, MO, USA) at 37 °C for 60 min before labeling.

### 2.5. Scrotal Ultrasound Evaluation

TV was evaluated by scrotal ultrasound. The ultrasound examination was performed with a GX MegasEsaote (EsaoteSpA, Genoa, Italy) device, equipped with linear, high-resolution, and high-frequency (7.5 to 14 MHz) probes dedicated to the study of soft body areas, with color Doppler for detecting slow flow and a scanning surface of at least 5 cm. The TV was calculated using the ellipsoid formula (length × width × thickness × 0.52). The testis was considered normal in size when it had a volume between 15 and 25 cm^3^, low normal when it had a volume between 10 and 12 cm^3^, and hypotrophic when it had a volume lower than 10 cm^3^ [12,13]. TV was calculated as the mean value between the volumes of the right and left testes.

### 2.6. Statistical Analysis

Results are reported as mean ± SD throughout the study. The normal distribution of each variable was evaluated by using the Shapiro–Wilks test. Within-group differences were analyzed by one-way analysis of variance (ANOVA) followed by the Tukey–Kramer post hoc test. Statistical analysis was performed using MedCalc Software Ltd. (Ostend, Belgium) (Version 19.6–64 bit). A *p* value lower than 0.05 was considered statistically significant.

### 2.7. Ethical Approval

This study was conducted at the Division of Endocrinology, Metabolic Diseases and Nutrition of the University teaching hospital “G. Rodolico–San Marco”, University of Catania (Catania, Italy). The protocol was approved by the internal Institutional Review Board and informed written consent was obtained from each participant after a full explanation of the purpose and nature of all procedures used. The study has been conducted according to the principles expressed in the Declaration of Helsinki.

## 3. Results

The overtime anthropometric, hormonal values, testicular volume, conventional sperm parameters, and SDF rate in the 20 patients with non-operated varicocele are shown in Table 1. A significant increase of gonadotropins and a decrease of TT and TV were observed, indicating a negative, although subclinical, the impact of varicocele on both Sertoli and Leydig cell function over time. In particular, serum FSH levels were significantly higher at T2 and T3 compared to T0, and serum LH levels were higher at all the follow-up time-points compared to T0. Furthermore, LH at T2 was higher than T1, and, at T3, it was higher than both T1 and T2, showing a time-dependent increase. TT was significantly lower at all the time points compared with T0. TV was lower at T2 and T3 compared with T0, and at T3 compared with T1. Finally, conventional sperm parameters and SDF rate did not show any significant change compared with T0 (Table 1).

Interestingly, serum PTH levels showed an increasing trend over time and its serum levels increased compared to baseline at all the time points assessed in the Patient group. Furthermore, both the values at T2 and T3 were significantly higher than those at T1. These values were also significantly higher at T1, T2, and T3 compared to the Control group (Figure 1A). Serum levels of calcium were significantly higher in T3 vs. T0 in the patient group. Compared to controls, patients showed significantly higher calcium serum levels at each assessed time. Additionally, patients and controls did not show serum calcium levels falling within the range of hypercalcemia (Figure 1B). Serum 25(OH)D showed decreasing trend over time in the Patient group. In particular, its levels decreased at all the time points assessed compared with T0. Additionally, 25(OH)D values at T2 and T3 were significantly lower than values at T1. Patients also showed significantly lower 25(OH)D serum levels at each assessed time compared to the control group (Figure 1C). Since the BMI of the enrolled patients did not change over time (Table 1), the decline of serum 25(OH)D levels does not seem related to an increase in the fat mass. These results indicate that the patients included in this study were at risk for developing serum 25(OH)D deficiency and secondary hyperparathyroidism, possibly due to a reduction of the activity of the testicular 25-hydroxylase.

## 4. Discussion

Varicocele is diagnosed in 15% of men [1] and, according to its severity and the patients’ susceptibility, it can lead to subclinical or clinical dysfunction of both Sertoli and Leydig cells. This can result in impaired conventional sperm parameters and pregnancy outcomes, increased gonadotropin levels, low serum TT levels, and even reduced TV.

Scanty evidence is currently available on the effect of varicocele on testicular CYP2R1 enzyme, which hydroxylases vitamin D into 25(OH)D. To evaluate this issue, we enrolled 20 consecutive patients with III degrees bilateral varicocele (according to the Dubin and Amelar classification), normal sperm parameter, and no indication of varicocele repair (no infertility, no scrotal pain, no decrease of testicular volume). They underwent measurements of serum PTH, calcium, and 25(OH)D at enrollment and annually for three years. Serum gonadotropin and TT levels, conventional sperm parameters, SDF rate, and TV were also evaluated. A significant increase of PTH and a specular decrease in 25(OH)D levels was observed over time. Concomitantly, signs of subclinical Leydig and Sertoli cell dysfunction, such as an increase in gonadotropins and a decrease in TT and TV over time was found. No change in conventional sperm parameters and SDF rate was observed.

These results suggest a decline of testicular 25-hydroxylase activity in these patients that explains the decrease of serum 25(OH)D levels and the consequent secondary hyperparathyroidism. This interpretation is further supported by the increase in serum LH levels. Accordingly, the testicular CYP2R1 activity has been reported to be LH dependent [14]. Furthermore, the administration with human chorionic gonadotropin (hCG) which has an LH-like activity, is indeed able to increase serum 25(OH)D levels in patients with late-onset hypogonadism and hypovitaminosis D [14]. Therefore, the increase in LH likely indicates Leydig cell dysfunction that concerns not only its steroidogenic activity but also that of 25-hydroxylase.

The decline of 25(OH)D that we observed in these patients does not seem related to changes in body weight. Indeed, vitamin D, due to its lipophilic nature, is stored in the adipose tissue, and obese patients have a greater chance of hypovitaminosis D [9]. It is likely that the reduction in vitamin D levels over time cannot be ascribed to a change in body composition in the enrolled patients, since the BMI does not change and that the patients included were of normal weight.

The activity of testicular steroidogenic enzymes can be influenced by thermic damage caused by varicocele. In adult rats with experimentally induced varicocele, the activities of the 17α-hydroxylase, 17,20-desmolase, and 17ß-hydroxysteroid dehydrogenase enzymes (involved all in testosterone biosynthesis) decrease significantly [15]. Likewise, a negative impact of varicocele also on the activity of the 25-hydroxylase may be hypothesized.

The results of the present study suggest, for the first time, that the management of bilateral, grade III, varicocele should include not only fertility and TT production but also the evaluation of PTH and Vitamin D. The long-term complications of bilateral varicocele may include, therefore, hypovitaminosis D and secondary hyperparathyroidism that, in turn, can affect bone mineralization, also in patients with no evidence of overt hypogonadism. However, there are no studies, so far, on the bone mineral density in patients with bilateral varicocele. In the next future, if further studies will confirm these results, indications for varicocele repair could include the onset of hypovitaminosis D and secondary hyperparathyroidism.

The strengths of this study include the prospective 3-year-long study design, the homogeneity of the cohort, the stringent inclusion criteria, and the presence of a control group. The main limitations of the study, in our opinion, are represented by lack of information regarding lower varicocele grades and inability to quantify the patients’ exposure time to varicocele before enrollment.

In conclusion, this prospective study provides the first evidence of a negative impact of bilateral grade III varicocele on the testicular 25-hydroxylase activity. Accordingly, the patients included in this study showed a significant increase in serum PTH and a parallel decrease in 25(OH)D levels over time. If further studies will confirm this evidence, endocrinologic counseling may be included in the management of varicocele.

## Figures and Tables

**Figure 1 jcm-11-00716-f001:**
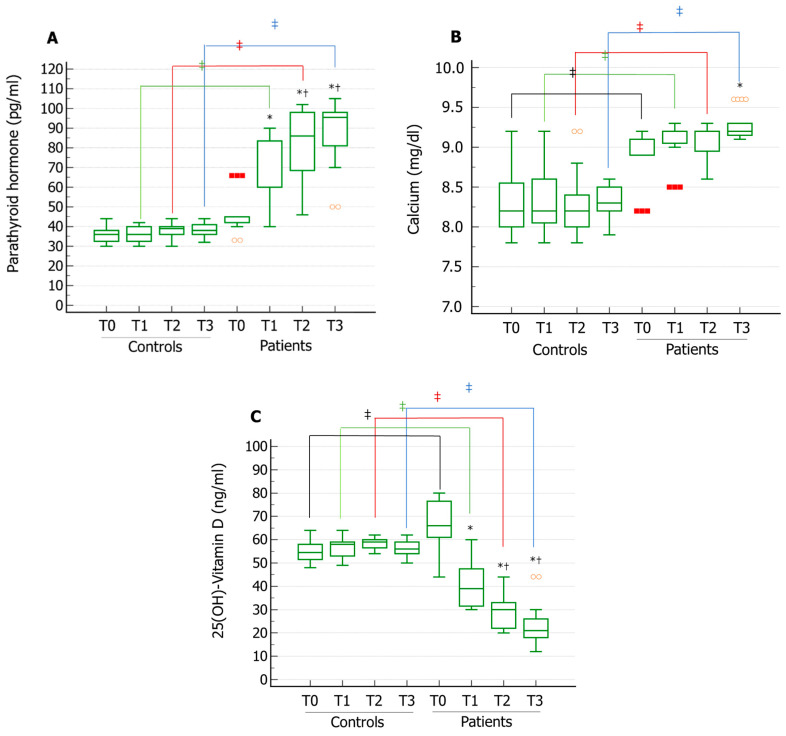
Serum levels of parathyroid hormone (PTH), calcium, and 25(OH)-Vitamin D in patients with III degree bilateral varicocele and varicocelectomized controls. Serum PTH levels significantly increased after one (T1), two (T2), and three (T3) years of follow-up compared to the values at enrolment (T0) in the patient group. Compared to controls, patients showed significantly higher PTH serum levels at T1, T2, and T3. (**A**). Serum calcium levels were significantly higher in T3 vs. T0 in the patient group. Compared to controls, patients showed significantly higher calcium serum levels at each assessed time. (**B**). Serum 25(OH)-Vitamin D levels significantly decreased over time in the patient group. Compared to controls, patients showed significantly lower 25(OH)-Vitamin D serum levels at each assessed time. (Panel (**C**)). * *p* < 0.05 vs. T0; ^†^ *p* < 0.05 vs. T1; ^‡^ *p* < 0.05.

**Table 1 jcm-11-00716-t001:** Trend over time of anthropometric, hormonal values, testicular volume, sperm parameters, and sperm DNA fragmentation in patients with III degrees bilateral varicocele.

Parameters	T0 (Mean ± SD)	T1 (Mean ± SD)	T2 (Mean ± SD)	T3 (Mean ± SD)
BMI (Kg/m^2^)	22.6 ± 1.4	22.9 ± 1.0	23.0 ± 0.8	22.8 ± 0.6
FSH (IU/L)	2.6 ± 0.4	3.0 ± 0.6	3.1 ± 0.5 *	3.2 ± 0.5 *
LH (IU/L)	2.3 ± 0.5	4.1 ± 0.7 *	5.5 ± 1.4 *^,†^	6.6 ± 1.5 *^,†,‡^
TT (ng/mL)	5.6 ± 0.4	4.9 ± 0.6 *	4.7 ± 0.6 *	4.5 ± 0.6 *
Total TV (mL)	16.8 ± 1.3	15.8 ± 1.6	15.1 ± 1.7 *	14.2 ± 1.7 *^,†^
Sperm concentration (million/mL)	78.3 ± 10.4	71.8 ± 8.9	78.5 ± 7.8	71.1 ± 9.5
Sperm progressive motility (%)	51.1 ± 7.1	46.6 ± 5.9	47.4 ± 6.1	51.2 ± 5.2
Spermatozoa with normal morphology (%)	13.0 ± 2.2	12.0 ± 1.5	11.4 ± 2.8	12.6 ± 1.7
SDF rate (%)	2.1 ± 2.5	1.8 ± 0.9	1.7 ± 0.9	1.9 ± 0.9

Abbreviations: BMI = body mass index; FSH = follicle-stimulating hormone; LH = luteinizing hormone; SD = standard deviation; SDF = sperm DNA fragmentation; T0 = baseline; T1 = 1-year follow-up; T2 = 2-year follow-up; T3 = 3-year follow-up; TT = total testosterone; TV = testicular volume. * *p* < 0.05 vs. T0; ^†^ *p* < 0.05 vs. T1; ^‡^ *p* < 0.05 vs. T2. Within-group differences were analyzed by one-way analysis of variance (ANOVA) followed by the Tukey-Kramer post hoc test.

## Data Availability

Data are available upon request to the corresponding author.

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
