# Peer review of "Is Chronic Varicocele a Risk Factor for Secondary Hyperparathyroidism?"

_jcm, 2022, doi:10.3390/jcm11030716_

Round 1

Reviewer 1 Report

In the manuscript "Is chronic varicocele a risk factor for secondary hyperparathy roidism?" the authors performed a prospective study provides the first evidence of a negative impact of bilateral grade III varicocele on the testicular 25-hydroxylase activity. The paper has certainly a clinical significance. However, some major criticisms are present, as follows:

- The authors do not present the baseline vitamin D, before follow-up, were the patients already presenting vitamin D deficiency

- The hypothesis is very interesting, but in the current situation, there is difficult to link the vitamin D deficiency to the varicocele

- Secondary hyperparathyroidism can occur in vitamin D deficiency, but that is not necessarily linked to the presence of varicocele

- The study has to have 2 cohorts, one with men with the same age and baseline vitamin D and PTH levels and compared it to the cohort with varicocele to see if there are any significant differences

- In the current form, I consider that the study is biased and the data is very unclear and reliable.

Author Response

Answers to Reviewer #1 comments

Manuscript ID jcm-1491605

Thank you for the comments and the time you spent in reviewing the present manuscript. We also appreciated your constructive criticisms.

Comment 1: The authors do not present the baseline vitamin D, before follow-up, were the patients already presenting vitamin D deficiency.

Answer to comment 1: The basal values of vitamin D are those at T0. We understand that this information was not written clearly in the Method section. So, we did this in the revised version of the manuscript (please see line 99. Regarding vitamin D deficiency, patients did not have vitamin D deficiency at T0 (please see Figure 1, panel C), as all values were above 40 ng/ml.

Comment 2: The hypothesis is very interesting, but in the current situation, there is difficult to link the vitamin D deficiency to the varicocele.

Answer to comment 2: To further support the hypothesis raised by the results of this study, we have added the data of age- and BMI-matched varicocelectomized control group. Differently from the non-varicocelectomized patients, they develop neither hyperparathyroidism nor vitamin D deficiency (please see the revised Figure 1). Finally, patients and controls did not have serum calcium levels compatible with the diagnosis of hypercalcemia (Figure 1). 

Comment 3: Secondary hyperparathyroidism can occur in vitamin D deficiency, but that is not necessarily linked to the presence of varicocele.

Answer to comment 3: Please see the answer to comment 2.

Comment 4: The study has to have 2 cohorts, one with men with the same age and baseline vitamin D and PTH levels and compared it to the cohort with varicocele to see if there are any significant differences.

Answer to comment 4: Done as requested. Thank you for this valuable comment.

Comment 5: In the current form, I consider that the study is biased and the data is very unclear and reliable.

Answer to comment 5: We hope that the addition of a control group improved the strength of the evidence provided by the present manuscript.

Reviewer 2 Report

In the manuscript, the authors discuss the potential influence of chronic varicocele on the activity of testicular 25-hydroxylase in Leydig cells and its influence on calcium-phosphate metabolism.

The issue is interesting and has not yet been widely assessed.

The main shortage of the study is the lack of a control group that would show the real effect of chronic varicocele on vitamin D deficiency.

There are also some minor issues that need clarification:

- Did the observation patients have any clinical symptoms of varicoceles?

- How long was the time between the onset of symptoms of varicocele and the inclusion in the observation?

- Is there information in the literature what percentage of vitamin D hydroxylation depends on the activity of Leydig cells' 25-hydroxylase?

- Are patients at risk of vitamin D deficiency? (e.g. due to the risk of renal failure)

- Have they supplemented with vitamin D in accordance with the general clinical practice guidelines of the endocrinology society?

- Were vitamin D and PTH measurements taken in the same season?

- Could you add PTH, Calcium and Serum 25 (OH) Vitamin D values ​​to Chart 1?

Author Response

Answers to Reviewer #2 comments

Manuscript ID jcm-1491605

Thank you for the comments and the time you spent in reviewing the present manuscript. We also appreciated your constructive criticisms.

Comment 1:  The main shortage of the study is the lack of a control group that would show the real effect of chronic varicocele on vitamin D deficiency.

Answer to comment 1: As suggested, we have added the data of age- and BMI-matched varicocelectomized control group. Differently from the non-varicocelectomized patients, they develop neither hyperparathyroidism nor vitamin D deficiency (please see the revised Figure 1).

Comment 2: Did the observation patients have any clinical symptoms of varicoceles?

Answer to comment 2: No, none of them had symptoms of varicocele (please see line 90).

Comment 3: How long was the time between the onset of symptoms of varicocele and the inclusion in the observation?

Answer to comment 3: They had no symptoms. We enrolled only patients with bilateral stage III varicocele. Thus, its onset was not recent. 

Comment 4: Is there information in the literature what percentage of vitamin D hydroxylation depends on the activity of Leydig cells' 25-hydroxylase?

Answer to comment 4: In healthy human testicular tissue, the expression of the CYP2R1 enzyme is known to be significantly higher compared to the liver (Foresta et al., 2011). Please see the following Figure taken from the study by Foresta and co-workers. We hope this answered satisfactorily your question.

Foresta C, Strapazzon G, De Toni L, Perilli L, Di Mambro A, Muciaccia B, Sartori L, Selice R. Bone mineral density and testicular failure: evidence for a role of vitamin D 25-hydroxylase in human testis. J Clin Endocrinol Metab. 2011 Apr;96(4):E646-52. doi: 10.1210/jc.2010-1628. Epub 2011 Jan 26. PMID: 21270327.

Comment 5: Are patients at risk of vitamin D deficiency? (e.g. due to the risk of renal failure).

Answer to comment 5: The patients enrolled in this study were all healthy. They did not have comorbidities or other risk factors of hypovitaminosis D (renal failure, obesity, malabsorption, lower exposure to sunlight –all of them live permanently in Sicily, the most Southern part of Italy) (lines 91-94).

Comment 6: Have they supplemented with vitamin D in accordance with the general clinical practice guidelines of the endocrinology society?

Answer to comment 6: No, they did not receive any vitamin D supplementation. This would have biased the results of the present study (lines 94-95).

Comment 7: Were vitamin D and PTH measurements taken in the same season?

Answer to comment 7: Yes, they were.

Comment 8: Could you add PTH, Calcium and Serum 25 (OH) Vitamin D values to Chart 1?

Answer to comment 8: Below you please find a table with the values of PTH, calcium, and serum 25(OH)-Vitamin D in patients and controls. We feel that adding the numerical values to the box plot chart will lower the figure quality. So, we prefer avoiding adding them.

Patients

Controls

PTH

Calcium

Vitamin D

PTH

Calcium

Vitamin D

T0

46.1±9.4

8.9±0.3

67.0±11.2

35.7±3.8

8.3±0.4

55.1±4.3

T1

67.7±16.3

9.1±0.3

41.0±10.8

36.0±3.8

8.3±0.4

56.5±4.1

T2

82.4±18.8

9.1±0.2

29.2±7.0

38.4±3.1

8.3±0.4

58.7±2.5

T3

88.7±17.0

9.3±0.2

23.0±8.7

38.1±3.6

8.3±0.2

56.5±3.1

Round 2

Reviewer 1 Report

Even if there is not clear connection between the disease, there is novelty and a general interest and further research could have clinical improvement.